# Manipulating Reaction Energy Coordinate Landscape of Mechanochemical Diaza-Cope Rearrangement

**DOI:** 10.3390/molecules27082570

**Published:** 2022-04-15

**Authors:** Tingting Cheng, Wenxian Ma, Hao Luo, Yangzhi Ye, KaKing Yan

**Affiliations:** 1School of Physical Science and Technology, ShanghaiTech University, Shanghai 201210, China; chengtt@shanghaitech.edu.cn (T.C.); mawx@shanghaitech.edu.cn (W.M.); luohao1@shanghaitech.edu.cn (H.L.); yeyzh@shanghaitech.edu.cn (Y.Y.); 2Shanghai Advanced Research Institute, Chinese Academy of Sciences, Shanghai 201203, China; 3University of Chinese Academy of Sciences, Beijing 100049, China

**Keywords:** ball mill, mechanochemistry, solid-state reactions, chiral vicinal diamine, silica gel

## Abstract

Chiral vicinal diamines, a unique class of optically-active building blocks, play a crucial role in material design, pharmaceutical, and catalysis. Traditionally, their syntheses are all solvent-based approaches, which make organic solvent an indispensable part of their production. As part of our program aiming to develop chemical processes with reduced carbon footprints, we recently reported a highly practical and environmentally-friendly synthetic route to chiral vicinal diamines by solvent-free mechanochemical diaza-Cope rearrangement. We herein showed that a new protocol by co-milling with common laboratory solid additives, such as silica gel, can significantly enhance the efficiency of the reaction, compared to reactions in the absence of additives. One possible explanation is the Lewis acidic nature of additives that accelerates a key Schiff base formation step. Reaction monitoring experiments tracing all the reaction species, including reactants, intermediates, and product, suggested that the reaction profile is distinctly different from ball-milling reactions without additives. Collectively, this work demonstrated that additive effect is a powerful tool to manipulate a reaction pathway in mechanochemical diazo-Cope rearrangement pathway, and this is expected to find broad interest in organic synthesis using mechanical force as an energy input.

## 1. Introduction

As an alternative synthetic approach, chemical reactions under the influence of mechanical force have experienced a renaissance in the synthetic organic community [1]. Although mechanochemistry was developed more than a century ago, it was not considered as a mainstream approach in organic synthesis until very recently. This gain of popularity stems partially from the fact that environmental friendliness in chemical synthesis is receiving more scientific consideration and general public attention [2]. Comparing to traditional solvent-based synthetic methods, ball-milling reactions require much less or, in a lot of cases, no organic solvent at all, representing an environmental benign and cost-effective and practical method [3,4,5,6,7]. Second, and more importantly, it often allows access to a unique chemical space that shows reactivity distinct from solution-based reactions [8,9,10,11,12,13,14,15,16,17,18,19,20,21,22,23,24]. There are ample examples in literature reporting dramatic rate acceleration of reactions performed under mechanochemical condition. The origin of rate enhancement could be due to (1) an increase in reaction concentration in the absence of solvent if the reactants are liquids; or (2) the in-situ formation of solid-state closed-packing microenvironment during ball-milling that favors reactions to take place [20,21].

Chiral vicinal diamines are important building blocks to a wide variety of pharmaceutical drug candidates, natural products, chiral ligands and catalysts in asymmetric synthesis, and both organic and metal-coordination porous materials [25,26,27,28,29,30,31,32,33,34,35,36,37,38,39,40,41,42,43,44,45,46,47,48,49,50,51]. Although they are widely applied in all aspect of chemical sciences, the existing methods to produce them are far from both practically and environmentally sustainable [52,53,54,55,56,57,58,59,60,61,62]. For example, most reported approaches gave racemic and even meso product mixtures (Figure 1a) [63,64,65,66]. As a result, a labor-intensive and multistep chiral separation or resolution is then needed to obtain optically pure chiral diamines. To overcome their limitations, both Chin [67,68] and Tang [69] independently developed new approaches revolving around [3,3]-sigmatropic rearrangement with chiral precursors through a two-step sequence to give chiral vicinal diamines (Figure 1b). These stereospecific methods are suitable for a broad scope of aryl aldehydes, and even ketones to give chiral diamines with quaternary stereogenic centers. Nevertheless, these powerful methods are all solution-based approaches, where the use of organic solvents is a pre-requisite of carrying out the transformations.

To make chiral diamine synthesis more environmentally friendly, we recently reported a solvent-free method to produce chiral vicinal diamines (Figure 1c) [70]. Building upon Chin’s work on resonance-assisted hydrogen bond (RAHB)-driven diaza-Cope rearrangement (DCR), we showed that a solvent-free mechanochemical method is much more effective compared to a conventional solution-based approach. Furthermore, during solid-state ball-milling, in-situ formation of solid-state packing effectively facilitates DCR [20,21]. Using this solvent-free method, we successfully synthesized a wide range of Schiff-based complexes including a sterically crowded benzocrown ether-containing Schiff-based ligand for allosteric chirality switching with alkanediammonium guests [71].

In general, conventional strategies to improve a mechanochemical reaction involve adjusting the milling frequency, time, and even milling temperature [72]. Other less subtle approaches such as changing the milling jars and balls into different materials (such as ZrO_2_, stainless steel, Teflon) and sizes also prove to be very effective. Recently, liquid-assisted grinding (LAG) has gained popularity in the synthetic community [72,73,74]. In LAG, a small amount of liquid is added to the reaction mixture during the ball-milling process to mainly promote better substrate mixing. Interestingly, LAG can even inhibit aggregation of molecular catalysts, as demonstrated by Ito in mechanochemically-driven Pd-catalyzed cross coupling reaction with a small amount of cyclohexene [75]. On the other hand, examination of solid additive is less explored. Herein, we show that a new protocol by co-milling with common laboratory solid additives, such as silica gel, can significantly enhance the efficiency of the reaction, compared to reactions in the absence of additives (Figure 1d). From a fundamental perspective, kinetic study suggested that the reaction pathway is distinct from both solvent-free reaction (without solid additives) and solution-based reaction, highlighting the importance of substrate–additive interaction to manipulate reaction pathways in mechanochemical DCR pathways. From a practical perspective, this work demonstrated that simple additives could find a pronounced effect in rate enhancement, and this is expected to find broad interest in organic synthesis using mechanical force as an energy input.

## 2. Results and Discussion

### 2.1. Solid Additive Effect in DCR Reaction between **1** and **2a**

#### 2.1.1. Screening of Different Common Laboratory Solid Additives for DCR Reaction

Addition of common laboratory solid auxiliary was shown in the past as a viable strategy to enhance reactivity in mechanochemical reactions. For example, Friščić and co-workers observed a drastic improvement in mechanochemical cross-metathesis reaction with sodium and potassium salts [76]. We commenced our study by examining different common laboratory solids as co-milling agents in a mechanochemical DCR reaction between (1*R*, 2*R*)-1,2-bis(hydroxyphenyl)ethylenediamine (chiral mother diamine) (**1**) and 4-(dimethylamino)benzaldehyde (**2a**) with a mixer mill machine. In a typical ball-milling reaction setup, **1** (0.041 mmol) and a slight excess of **2a** (0.087 mmol, 2.1 equiv.) were loaded into a 2 mL PP (PP = polypropylene) centrifuge tube equipped with ZrO_2_ milling balls (3 mm × 5). After ball-milling, a milled sample was dissolved in DMSO-*d*_6_ and the NMR sample was injected into an NMR spectrometer as quickly as possible (within 2 min). After that, data acquisition began immediately to acquire the ^1^H NMR spectrum. Although we have shown that solution based DCR is much slower than the corresponding ball-milling reaction, our protocol minimizes further potential solution reaction of the solid sample after prolong period in solution. At 40 Hz for 30 min, in the absence of any solid additive, the reaction gave a DCR product **3a** in 18% yield, detected by ^1^H NMR spectroscopy using an internal standard (Table 1, entry 1). Addition of NaCl (20 mg) does not facilitate the formation of **3a** with <5% yield (Table 1, entry 2). Although it was known that Schiff-base formation, a key first step in DCR, could be catalyzed by both base and acid catalysts, after screening a wide range of solid basic reagents, including various alkali metal carbonates, acetates, etc., none of them improve the yield of **3a** (Table 1, entry 3–11), nor did organic acids. For instance, the addition of either TsOH or B(OH)_3_ as a co-milling agent completely shut down the reaction (Table 1, entry 12–13). No sign of any DCR product **3a** was detected. To our surprise, AlCl_3_ (Table 1, entry 14), reported to catalyze Schiff base formation [77], failed to promote DCR. When a mixture of **1** and **2a** was ball-milled in the presence of silica gel, we were pleased to find that **3a** was formed in a satisfactory 62% yield after 30 min (Table 1, entry 19). Remarkably, replacing silica gel with sand, a crystalline and nonporous form of SiO_2_, proved to be ineffective for DCR (Table 1, entry 20). Finally, as iron could be present as a trace impurity in silica gel, we also evaluated the effectiveness with iron salt as a co-milling reagent. However, only 9% yield was achieved with Fe_2_O_3_ (Table 1, entry 21), totally eliminating the possibility of trace impurity catalyzing DCR.

#### 2.1.2. Screening of Silica Gel Loading for DCR Reaction

After identifying silica gel as an effective catalyst for DCR, we next studied the effect of silica gel loading, from 10 to 100 mg (Appendix A). With 10 mg of silica gel as co-milling agent in our typical ball-milling condition using **1** (0.082 mmol) and **2a** (0.174 mmol), a respectable 48% yield of **3a** could be detected by ^1^H NMR spectroscopy (Appendix A, entry 2). Increasing the silica gel amount to 20 mg improves the yield of **3a** to 59% (Appendix A, entry 3). Further increase in silica gel loading leads to a steady increase in yield of **3a** to 75% with 50–100 mg of silica gel (Appendix A, entry 5–6). For NMR scale reaction, DMSO-*d*_6_ was added to the milled sample directly. However, the workup (filtration) becomes difficult with too much silica gel additive, as loss of product would underestimate the reaction yield. Therefore, reaction with 20 mg of silica gel represents a balance of catalytic performance and easy workup. Therefore, for ease of ^1^H NMR sample preparation, we decided to conduct further studies with the use 20 mg of silica gel.

#### 2.1.3. Comparison of Different Reaction Methods for DCR Reaction

We next attempted to monitor the reaction time courses and study the kinetic profiles of solvent-free ball-milling reactions with and without silica gel additive and compare them to conventional solution reaction in DMSO-*d*_6_ we reported recently [70]. By measuring the conversion to **3a** from **1** and **2a** for the three strategies, we would have a more general perspective into the rate enhancement induced by silica gel. Conventional additive-free ball-milling gave **3a** in 18% after 30 min and 29% after 60 min (Figure 2, red trace). The conversion to **3a** reached 66% after milling for 4 h. Meanwhile, the corresponding reaction in the presence of silica gel (20 mg) gave **3a** in 37% after just 5 min (Figure 2, blue trace). Notably, there was a slight decrease in rate after 10 min of milling, and we contributed that to uneven mixing in a small milling vessel. Even that, the conversion eventually reached 68% yield after 60 min. Meanwhile, solution synthesis in DMSO-*d*_6_ gave **3a** in only 4% after almost 4h at rt (Figure 2, green trace).

#### 2.1.4. Investigation of the Role of Silica Gel in DCR Reaction

Moreover, the influence of silica gel particle sizes was further assessed. However, the results of DCR reaction with different meshes of silica gel were all comparable. That is, mechanochemical reaction using silica gel with various meshes gave rearranged product **3a** in 50–62% yield at 40 Hz for 30 min (Table 2, entry 1–4), showing the generality of silica gel as a unique catalyst for DCR reaction. 

Based on silica gel-catalyzed examples reported in prior literature, its catalytic effect stems from the silanol groups dispersed on the surface of silica gel behaving as Brønsted acid catalysts [78,79,80,81]. Therefore, we anticipated that thermal treatment of silica would lead to condensation of neighboring silanol groups on surface to give siloxane linkages between 200 to 750 °C [82,83]. The reduced concentration in silanol groups of activated silica material is expected to show reduced catalytic activity in DCR reaction. In order to test this hypothesis, the silica gel sample (300–400 mesh) used in entry 4 in Table 2 was subjected to thermal treatment at either 314 °C or 528 °C under vacuum for 5 h to give activated silica gel samples, **SG**_314_ and **SG**_528_, respectively. FT-IR spectra of **SG**_(314)_ and **SG**_(528)_ revealed an almost flat band in the OH stretching region (~2800–3500 cm^−1^) (Appendix A). These activated samples were then applied to mechanochemical DCR reactions with **1** and **2a**. ^1^H NMR spectra acquired immediately after ball-milling for 30 min at 40 Hz using **SG**_314_ and **SG**_528_ gave **3a** in 20% and 16%, respectively, supporting the collapse of silanol groups at high temperature (Table 2, entry 5–6), and reduction of the number of catalytic active sites for DCR reactions. 

In addition, we examined DCR reaction employing structurally-analogous materials such as silicon dioxide (Table 1, entry 20), alumina (Table 1, entry 18), and other metal oxides, such as CeO_2_, La_2_O_3_, and ZrO_2_ (Table 1, entry 15–17). However, these materials exhibit modest catalytic activity, further emphasizing the synergetic effect of hydroxyl groups in silica material. Upon high frequency collision between aldehyde substrates **2** and mother diamine **1** with the silanol active site under ball-milling condition, this leads to a greatly enhanced reaction rate observed for DCR reaction.

### 2.2. Scope Comparison for Ball-Milling-Induced DCR Reactions with and without Silica Gel Additive

To highlight the effect of silica gel in catalyzing DCR reaction, we also examined aldehyde substrates that have low conversion under our previously reported solvent-free mechanochemical condition, and the results are shown in Figure 3. For example, electron-withdrawing substrates **2b** and **2c** would give rearrangement products **3b** and **3c** in 34% and 20% yield, respectively, facilitated by mechanical ball-milling at 40 Hz for 30 min without solid additive. Under the new protocol with 20 mg of silica gel, the conversion to **3b** and **3c** can be boosted up to 70% and 95% yield, respectively. Likewise, *p*-phenylbenzaldehyde (**2d**) and *o*-iodobenzaldehyde (**2e**), sluggish substrates that require long ball-milling time and larger milling vessel and milling balls, give DCR products in moderate 41% and 71% yield, respectively, with a silica gel co-milling agent. Likewise, formation of *o*-phenyl and bithiophenyl products **3f** and **3g** can be improved to 34% and 83% yield, respectively, after just 30 min of ball-milling at 40 Hz with **1** and silica gel, while the reaction without silica gel would barely give any DCR products (<5% yield) under the same milling condition. Finally, other substrates (**2h**–**2i**) that were not reported in the past using DCR reactions with mother diamine **1** can also be transformed into the corresponding diimine species **3h–3i** in good yield, including one with sterically encumbered bis(3-bromonapthalenyl) substituent **3i**. Furthermore, the enantioselectivity of the resulting product (*S*,*S*)-**3** was confirmed by HPLC with chiral stationary to be >95% ee.

### 2.3. Mechanistic Studies of Silica Gel Catalyzed Mechanochemical DCR Reaction

#### 2.3.1. Tracking the Reaction Profile in DCR Reaction with Silica Gel Additive by ^1^H NMR Spectroscopy

The observed rate enhancement with silica gel is worthy of further investigation. When the kinetics profile was examined more closely, namely, tracking the concentration change for each observable species in mechanochemical reaction between **1** and **2a**, we could potentially differentiate the mechanistic variation between reaction methods shown in Figure 2 (ball-milling with and without silica gel). At each time point in Figure 4, we determined the concentration (or %) of mother diamine **1**, formation of five-membered imidazolidine intermediate (**4**), diimine before DCR (**5**), and imine product **3a** after DCR, with an internal standard added to the NMR sample. As periodic reaction tracking requires stopping the ball-mill reaction and subjecting the milled sample to solution NMR analysis, each time point was acquired from an independent experiment. In the reaction time course shown in Figure 4, the concentration of mother diamine **1** reduced rapidly to below 50%, together with a simultaneous spike of **3a** formation in the first 5 min of ball-milling. The major mechanistic difference for DCR reaction with silica gel is the consistently low concentration of **4** during the course of reaction. On the contrary, DCR reaction in the absence of silica gel would build up significant concentration of imidazolidine intermediate **4** [70]. This phenomenon suggests that silica gel catalyzes the conversion from **4** to **5**. As **4** was previously proposed as the longest-lived intermediate in mechanochemical DCR (without silica gel), disfavoring the formation of **4** from **6** by silica gel would drastically increase the rate of the overall reaction [70].

#### 2.3.2. Mechanistic Proposal of Mechanochemical DCR Catalyzed by Silica Gel Additive

In the absence of silica gel, a well-ordered crystalline phase was previously detected by powder X-ray diffraction (PXRD) during reaction [70]. We proposed that this progressive structural ordering phenomenon (in-situ crystallization) [20,21] would drive DCR reaction even when milling was stopped short of full conversion. When a silica gel-containing reaction between **1** and **2a** was ball-milled for 5 min at 40 Hz, ^1^H NMR spectrum revealed the formation of **3a** in 24% yield. This sample was next subjected to room temperature aging (free-standing with no milling), and the yield of **3a** reached 63% after 7 d. A broad peak between 15° and 28° in all milled silica gel-containing samples (before and after aging) in PXRD spectra was observed and was assigned to disordered phase, amorphous silica (Appendix A) [84,85]. This result clearly shows that the origins of DCR reaction rate enhancement with or without silica gel are distinctly different, and we attributed the driving force in the presence of silica gel to substrate activation by a silanol group on silica gel under a solvent-free condition.

By establishing the importance of silanol groups on silica gel shown above and the information gathered in the time course tracing experiments, we tentatively proposed a mechanism to describe the silica gel-induced DCR, outlined in Figure 5. Upon hydrogen bonding interaction with silanol groups, aldehyde substrate **2** is activated for nucleophilic attack by mother amine **1** to form mono-Schiff base intermediate **6**, which converts rapidly to five-membered imidazolidine **4** in the absence of silica gel. With silica gel, this process is disfavored. Instead, **6** engages a second Schiff base formation with activated **2** to give diimine intermediate **5**. A similar type of hydrogen bonding interaction between silanol and hydroxyl group on **5** would further stabilize and rigidify the six-membered ring transition state required for DCR to give final product **3**. Activation of substrate and intermediates by surface silanol groups is expected to reduce the energy barriers of all the steps in the reaction mechanism, and thus results in the observed rate acceleration. This also explains why intermediate **6** was scarcely detected during reaction tracing in the presence of silica gel (Figure 5), a sharp contrast to reaction without silica gel. 

### 2.4. The Effect of Liquid Additives in DCR Reaction by Liquid-Assisted Grinding (LAG)

Finally, the influence of liquid in DCR reaction was explored as a direct comparison to solid additives. Liquid-assisted grinding (LAG) is gaining much recognition as an alternative mechanochemical strategy applied in organic synthesis, production of pharmaceutical, and functional materials [86,87,88,89]. In LAG, trace amount of liquids is added to the milling mixture to facilitate better mixing efficiency. For example, Ito and others reported numerous examples in mechanochemical cross-coupling reactions with dramatic reaction rate acceleration by the LAG method [75,90,91,92,93]. Browne [23] and Mack [94] independently showed that LAG could even switch the selectivity outcome of chemical reactions to bias the formation of thermodynamically unfavorable kinetic products.

Ethanol and DMSO were previously reported by Chin as preferred solvent media for DCR reactions [67]. By employing EtOH (dielectric constant (ε) = 24.5) by LAG at 40 Hz for 30 min, we showed that the reactions were sluggish (<5% yield in all cases) regardless of the amount of EtOH applied (from 0.5–2.0 µL/mg) (Table 3, entry 1–4), although the reactants (**1** and **2a**) are partially soluble under reaction conditions. Switching to DMSO (ε = 46.7) (0.5 µL/mg) improves the formation of **3a** to 25% yield (Table 3, entry 5) with enhanced solubility of reactants. Interestingly, when DCR reaction of **1** and **2a** was conducted in a slightly polar solvent in DCM (ε = 8.9), the yield of **3a** reached 54% yield (Table 3, entry 6). This implies that solubility of reaction components is not the dominant factor to effective transformation. Surprisingly, LAG with non-polar solvent in petroleum ether, where reactants are completely insoluble, proved to be ineffective to promote DCR reaction (<5% yield) (Table 3, entry 7). Considering the environmental impact of organic solvent, especially halogenated solvents, DCR reaction with solid additive could be a superior choice compared to employing liquids. 

## 3. Materials and Methods

### 3.1. General Information

All commercial reagents and solvents were used as received unless otherwise indicated. (1*R*,2*R*)-1,2-Bis(2-hydroxyphenyl)ethylenediamine and (1*S*,2*S*)-1,2-Bis(2-hydroxyphenyl)ethylenediamine was purchased from Sigma Aldrich. Other reagents and solvents were purchased from Sinopharm Chemical Reagent, Bidepharm, and J&K Chemical. Deuterated solvents were acquired from Adamas, and Eurisotop. Chromatographic purification of products was accomplished by flash chromatography using silica gel. Thin-layer chromatography (TLC) was performed on ShanXi NuoTai silica gel SHF254 TLC plates. NMR spectra were obtained from Bruker Avance NEO 400 and Bruker Avance III HD 500. ^1^H NMR spectra were recorded at 500 MHz or 400 MHz, and ^13^C{^1^H} NMR spectra were recorded at 125 MHz or 100MHz. ^1^H NMR chemical shifts were determined relative to the signal of the residual protonated solvent. ^11^B NMR NMR spectrum was recorded at 128 MHz. Data for ^1^H NMR are reported as follows: chemical shift (δ ppm), multiplicity (s = singlet, d = doublet, t = triplet, m = multiplet). ^13^C{^1^H} NMR chemical shifts of NMR spectra were reported as chemical shifts in ppm and multiplicity where appropriate. High Resolution Mass spectra were obtained from Thermo Fisher Q-Exactive high-resolution MS, operated in ESI mode. Enantiomeric excesses were determined on an Agilent 1260 Chiral HPLC using IC, and IA columns. FT-IR data were recorded on a Thermo Scientific Nicolet iS5 FTIR spectrometer. 

Diaza-Cope products **3b**–**3d** were previously reported and characterized [95,96]. The analytical data in Figure 3 matches those of the literature.

### 3.2. General Procedure for Mechanochemical Diaza-Cope Rearrangement (0.04 mmol Scale)

(1*R*,2*R*)-1,2-Bis(2-hydroxyphenyl)ethylenediamine **1**, arylaldehyde **2** (2.1 equiv.), and silica gel (20 mg) were placed in a polypropylene ball-milling vessel (2 mL) loaded with five grinding ball (ZrO_2_, diameter: 3 mm). After the vessel was closed in open air, the vessel was placed in the ball mill machine. After 30 min of ball-milling at 40 Hz, DMSO-*d*_6_ was added to the vessel and filtered out the silica gel. The filtrate was injected into an NMR tube with a known amount of internal standard (dibromomethane), and the NMR sample was measured by NMR spectroscopy to determine the conversion to DCR product **3**. 

### 3.3. General Procedure for Mechanochemical Diaza-Cope Rearrangement (0.2 mmol Scale)

When the reactions are scaled up 5X from 0.04 mmol to 0.2 mmol, the amount of silica gel was also increased from 20 mg to 100 mg, which would be too much to pack in a 2 mL or 5 mL PP milling vessel. For PP tubes with volume ≥ 10 mL manufactured for the mixer mill machine used in this study, their shapes are long and narrow, which are not suitable for well-mixing. With the equipment limitation, we decided to do the reactions in a 15 mL stainless-steel (SS) vessel with stainless steel milling ball instead. However, as the control experiment in Table 1, entry 19 indicated, SS vessel is less suitable for DCR reaction in the presence of silica gel. This hampered the highest possible yield that we can achieve. A sample protocol for 0.2 mmol scale DCR reaction is listed below:

(1*R*,2*R*)-1,2-Bis(2-hydroxyphenyl)ethylenediamine **1**, arylaldehyde **2** (2.1 equiv.), and silica gel (100 mg) were placed in a stainless-steel ball-milling vessel (15 mL) loaded with one grinding ball (stainless steel, diameter: 12 mm). After the vessel was closed in open air, the vessel was placed in the ball mill machine. After 4 h of ball-milling at 40 Hz, with a periodic 5 min pause for every 1 h of milling, the solid residual was dissolved in DCM (10 mL), followed by purification by column chromatography (hexane/ethyl acetate = 10/1) to give product **3**.

### 3.4. Characterization Data for Products

2,2’-[[(1*S*,2*S*)-1,2-di(2-iodophenyl)-1,2-ethanediyl]bis[(*E*)-nitrilomethylidyne]]bisphenol (**3e**). Prepared according to general procedure from **1** (50 mg, 0.20 mmol) and 2-iodobenzaldehyde (**2e**) (98 mg, 0.42 mmol) to give the title compound as a yellow solid (82 mg, 61%). ^1^H NMR (CDCl_3_, 400 MHz, 25 °C) δ 13.06 (s, 2H), 8.44 (s, 2H), 7.69 (ddd, *J* = 15.6, 7.9, 1.5 Hz, 4H), 7.33–7.26 (m, 4H), 7.19 (dd, *J* = 7.7, 1.7 Hz, 2H), 6.94–6.91 (m, 2H), 6.90–6.85 (m, 2H), 6.82 (td, *J* = 7.4, 1.1 Hz, 2H), 5.44 (s, 2H). ^13^C{^1^H} NMR (CDCl_3_, 101 MHz, 25 °C) δ 166.73, 160.90, 140.61, 139.68, 132.82, 132.05, 130.68, 129.64, 128.67, 118.89, 118.53, 116.93, 100.52, 80.33. HRMS (ESI) calculated for C28H22I2N2O2, [M+H]^+^: 672.9843, Found: 672.9843. 

2,2’-[[(1*S*,2*S*)-1,2-di((2-phenyl)phenyl)-1,2-ethanediyl]bis[(*E*)-nitrilomethylidyne]]bisphenol (**3f**). Prepared according to general procedure from **1** (50 mg, 0.20 mmol) and [1,1’-biphenyl]-2-carbaldehyde (**2f**) (78 mg, 0.42 mmol) to give the title compound as a yellow solid (78 mg, 68%). ^1^H NMR (CDCl_3_, 400 MHz, 25 °C) δ 13.05 (s, 2H), 8.00 (s, 2H), 7.23 (dt, *J* = 7.1, 4.0 Hz, 4H), 7.11–6.94 (m, 12H), 6.88–6.84 (m, 2H), 6.79–6.68 (m, 6H), 6.62 (td, *J* = 7.5, 1.1 Hz, 2H), 4.87 (s, 2H). ^13^C{^1^H} NMR (CDCl_3_, 101 MHz, 25 °C) δ 165.64, 160.82, 142.08, 140.68, 135.95, 132.49, 131.73, 129.93, 129.66, 128.67, 128.14, 128.03, 127.83, 127.29, 127.18, 127.00, 118.75, 118.56, 116.98, 116.77. HRMS (ESI) calculated for C40H32N2O2, [M+H]^+^: 573.2537, Found: 573.2544. 

2,2’-[[(1*S*,2*S*)-1,2-di(2-(4,4,5,5-tetramethyl-1,3,2-dioxaborolan-2-yl)phenyl)-1,2-ethanediyl]bis[(*E*)-nitrilomethylidyne]]bisphenol (**3h**). Prepared according to general procedure from **1** (50 mg, 0.20 mmol) and 2-(4,4,5,5-tetramethyl-1,3,2-dioxaborolan-2-yl)benzaldehyde (**2h**) (98 mg, 0.42 mmol) to give the title compound as a yellow solid (30 mg, 22%). ^1^H NMR (CDCl_3_, 400 MHz, 25 °C) δ 13.65 (s, 2H), 8.53 (s, 2H), 7.75 (d, *J* = 7.8 Hz, 2H), 7.58 (dd, *J* = 7.5, 1.5 Hz, 2H), 7.32 (td, *J* = 7.6, 1.5 Hz, 2H), 7.23 (ddd, *J* = 8.5, 7.2, 1.7 Hz, 2H), 7.17 (dd, *J* = 7.6, 1.7 Hz, 2H), 7.07 (td, *J* = 7.4, 1.1 Hz, 2H), 6.88 (d, *J* = 8.2 Hz, 2H), 6.79 (td, *J* = 7.5, 1.1 Hz, 2H), 6.12 (s, 2H), 1.45 (s, 12H), 1.38 (s, 12H). ^13^C{^1^H} NMR (CDCl_3_, 101 MHz, 25 °C) δ 165.89, 161.07, 145.97, 136.22, 132.15, 131.69, 131.15, 128.14, 126.49, 119.02, 118.52, 116.71, 83.81, 25.05. HRMS (ESI) calculated for C40H46B2N2O6, [M+H]^+^: 673.3615, Found: 673.3659. 

2,2’-[[(1*S*,2*S*)-1,2-di(3-bromonaphthalen-2-yl)-1,2-ethanediyl]bis[(*E*)-nitrilomethylidyne]]bisphenol (**3i**). Prepared according to general procedure from **1** (50 mg, 0.20 mmol) and 3-bromo-2-naphthaldehyde (**2i**) (99 mg, 0.42 mmol) to give the title compound as a yellow solid (76 mg, 48%). ^1^H NMR (CDCl_3_, 500 MHz, 25 °C) δ 13.28 (s, 2H), 8.42 (s, 2H), 8.26 (d, *J* = 8.6 Hz, 2H), 7.82 (d, *J* = 8.6 Hz, 2H), 7.70 (t, *J* = 8.1 Hz, 4H), 7.52 (t, *J* = 7.7 Hz, 2H), 7.45 (t, *J* = 7.5 Hz, 2H), 7.28 (t, *J* = 7.6 Hz, 2H), 7.13 (d, *J* = 7.6 Hz, 2H), 6.97 (d, *J* = 8.3 Hz, 2H), 6.80 (t, *J* = 7.5 Hz, 2H), 6.06 (s, 2H). ^13^C{^1^H} NMR (CDCl_3_, 101 MHz, 25 °C) δ 167.30, 161.03, 136.03, 133.98, 132.84, 132.13, 132.03, 128.15, 128.05, 128.00, 127.49, 126.85, 123.95, 118.84, 118.58, 117.02, 76.46. HRMS (ESI) calculated for C36H26Br2N2O2, [M+H]^+^: 679.0413, Found: 679.0427. 

## 4. Conclusions

In this work, diaza-Cope rearrangement reaction was examined under mechanochemical conditions in the presence of different additives. Among twenty solid additives tested, including various metal salts, inorganic bases, Lewis’s acids and Bronsted acids, silica gel gave the best conversion. Mechanistic studies, including kinetic tracing and “activated” silica gel experiments, suggested that silanol groups on silica gel are quintessential in enhancing both Schiff base formation and DCR by disfavoring imidazolidine intermediate **4** accumulation during reaction, which was previously shown as a long-lived intermediate in mechanochemical DCR without silica gel. Collectively, an incredibly efficient DCR reaction strategy was developed featuring no requirement of organic solvent for transformation, broad substrate scope and complete stereospecificity. As such, this environmentally benign method is expected to gather academic and industrial interest for potential chiral vicinal diamine synthesis, an important class of building blocks that plays an important role in material design, pharmaceutical, and catalysis.

## Data Availability

The data reported in this study are available in the Appendix A.

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
