# Peer review of "Manipulating Reaction Energy Coordinate Landscape of Mechanochemical Diaza-Cope Rearrangement"

_molecules, 2022, doi:10.3390/molecules27082570_

Round 1
Reviewer 1 Report
The manuscript by Yan et al deals with the study of the diaza-Cope rearrangement reaction under mechanochemical conditions on chiral diamines. The work is well written with clear experimental procedures. The study covers an interesting range of experimental conditions to try to understand the mode of action of the silica gel added as an additive in the reaction. In my opinion, the work is of interest to be published in Molecules after the authors attend to the following two suggestions:
1) Table 2 (in section 2.1.2) is not relevant to the understanding of the work and should be transferred to the supplementary material.
2) Section 2.1.3 (and scheme 2): The authors should justify why this section should remain in the work, given that several of these experiences were already presented in a previous publication by the same authors (reference 74). I suggest that the authors only present the new results or justify why they consider it necessary to reiterate already published data.
Author Response
Comment 1: The manuscript by Yan et al deals with the study of the diaza-Cope rearrangement reaction under mechanochemical conditions on chiral diamines. The work is well written with clear experimental procedures. The study covers an interesting range of experimental conditions to try to understand the mode of action of the silica gel added as an additive in the reaction.
Response 1: We sincerely thank Reviewer 1 for his/her time to review our manuscript, and thank for the high praise of our work.
Comment 2: 1) Table 2 (in section 2.1.2) is not relevant to the understanding of the work and should be transferred to the supplementary material.
Response 2: We have moved this table to the ESI section as Table S1.
Comment 3: 2) Section 2.1.3 (and scheme 2): The authors should justify why this section should remain in the work, given that several of these experiences were already presented in a previous publication by the same authors (reference 74). I suggest that the authors only present the new results or justify why they consider it necessary to reiterate already published data.
Response 3: We thank you for the reviewer’s comment. However, we think this scheme is extremely critical for the readers to get a visual idea of how our current method (ball-milling in the presence of silica gel) is superior to both simple ball-milling and solution-based reactions. As mentioned in the footnote in Scheme 2, only the solution-based reaction data was reported before in Ref 7. To show a better comparison, we modified the ball-milling (without silica gel) reaction condition and re-examined the kinetic study (Ref 7: 15 mL stainless steel jar, 12 mm stainless steel ball x 1 vs. this work: 2 mL polypropylene jar, 3 mm ZrO2 ball x 5). This reaction represents a new reaction demonstrated in this work for the first time. Therefore, I respectively disagree with the reviewer’s opinion to remove this section from the manuscript, and we ask to keep this section in the main text.
Reviewer 2 Report
Cheng et al in the manuscript "Manipulating Reaction Energy Coordinate Landscape of Mechanochemical Diaza-Cope Rearrangement" reported about a mechanochemical protocol diazo-Cope rearrangement using solid additive, as silica gel. They investigated different solid additive and they selected silica gel as the most active to produce the highest yield. Then they played mechanistic studies and suggested that silanol groups on silica gel activated the substrates leading to key base Schiff intermediate formation step.
I suggest to revise the manuscript as follows:
- in the section "2.1.4. Investigation of the role of silica gel in DCR reaction" the authors investigated the role of the size of silica gel particles on the DCR reaction. In the samples 5 and 6 in table 3 they treated the silica particles at two different temperatures. How they selected these conditions? please explain this choice.
- in the section "2.2. Scope comparison for ball-milling-induced DCR reactions with and without silica gel additive" in Scheme 3,the authors did not represent compounds 2b-i. For the sake of clearness, these structures should be inserted into the Scheme.
- the authors did not reported compond 3b-d characterization or bibliographic references for them.
Others comments:
- row 194 "In order to test this hypothesis" instead "In order test this hypothesis"
- row 199 is not clear "The diminishing DCR efficiency is correlated to the reduced concentration of silanol groups of these of activated silica samples.
- row 206 "silicon dioxide" instead of "silicon oxide"?
- row 237 "mother diamine 1 can also be transformed into the corresponding diimine species 3h-3i in good yield" instead of "mother diamine 1 can also transformed into the corresponding diimine species 3h-3i in good yield"
- row 278 "After aging (free standing with no milling) for 7 d at rt, the yield of 3a reached 63%." This sentence is not clear. 7 d at rt?
- row 310 "Liquid-assisted grinding (LAG) is gaining much recognition as an alternative mechanochemical strategy applied in organic synthesis, production of pharmaceutical, and functional materials." A reference is missing.
Author Response
Comment 1: Cheng et al in the manuscript "Manipulating Reaction Energy Coordinate Landscape of Mechanochemical Diaza-Cope Rearrangement" reported about a mechanochemical protocol diazo-Cope rearrangement using solid additive, as silica gel. They investigated different solid additive and they selected silica gel as the most active to produce the highest yield. Then they played mechanistic studies and suggested that silanol groups on silica gel activated the substrates leading to key base Schiff intermediate formation step.
Response 1: We again thank Reviewer 2 for his/her time to review our manuscript.
Comment 2: in the section "2.1.4. Investigation of the role of silica gel in DCR reaction" the authors investigated the role of the size of silica gel particles on the DCR reaction. In the samples 5 and 6 in table 3 they treated the silica particles at two different temperatures. How they selected these conditions? please explain this choice.
Response 2: We thank you for the reviewer’s comment. It was reported that thermal treatment would induce condensation of surface hydroxyl groups to form siloxane linkers on silica surfaces between 200 and 750 °C under vacuum. The main objective of the activated silica gel study in Table 2 is to show that if the number of silanol groups reduces, it would directly affect the efficiency of diazo-Cope rearrangement. Therefore, we chose the maximum temperature our in-house furnace can run (which is 528 °C) and a temperature that is below that but above 200 °C. We have added the corresponding citations in the reference section.
Comment 3: in the section "2.2. Scope comparison for ball-milling-induced DCR reactions with and without silica gel additive" in Scheme 3, the authors did not represent compounds 2b-i. For the sake of clearness, these structures should be inserted into the Scheme.
Response 3: We added the structure of 2b-i in Scheme 3.
Comment 4: the authors did not reported compond 3b-d characterization or bibliographic references for them.
Response 4: Characterizations of these compounds were previously reported, and their references are added to the reference section. A sentence was added to the General Information (Section 3.1):
“Diaza-Cope products 3b-3d were previously reported and characterized. The analytical data in Scheme 3 matches with those of literature.”
Comment 5: row 194 "In order to test this hypothesis" instead "In order test this hypothesis"
Response 5: This is fixed.
Comment 6: row 199 is not clear "The diminishing DCR efficiency is correlated to the reduced concentration of silanol groups of these of activated silica samples.
Response 6: We thank the reviewer to point this out. We decided to remove this sentence as we have already mentioned in line 192 that “The reduced concentration in silanol groups of activated silica material is expected to show reduced catalytic activity in DCR reaction. “, and we modified the paragraph to:
“FT-IR spectra of SG(314) and SG(528) revealed an almost flat band in the OH stretching region (~2800-3500 cm-1) (Figure S6). These activated samples were then applied to mechanochemical DCR reactions with 1 and 2a. 1H NMR spectra acquired immediately after ball-milling for 30 min at 40 Hz using SG314 and SG528 gave 3a in 20% and 16%, respectively, supporting the collapse of silanol groups at high temperature (Table 2, entry 5-6), and reduction of the number of catalytic active sites for DCR reactions. “
Comment 7: row 206 "silicon dioxide" instead of "silicon oxide"?
Response 7: This is fixed.
Comment 8: row 237 "mother diamine 1 can also be transformed into the corresponding diimine species 3h-3i in good yield" instead of "mother diamine 1 can also transformed into the corresponding diimine species 3h-3i in good yield"
Response 8: This is fixed.
Comment 9: row 278 "After aging (free standing with no milling) for 7 d at rt, the yield of 3a reached 63%." This sentence is not clear. 7 d at rt?
Response 9: We thank the reviewer for pointing this out. We modified the sentence.
“This sample was next subjected to room temperature aging (free-standing with no milling), and the yield of 3a reached 63% after 7 d.”
Comment 10: row 310 "Liquid-assisted grinding (LAG) is gaining much recognition as an alternative mechanochemical strategy applied in organic synthesis, production of pharmaceutical, and functional materials." A reference is missing.
Response 10: Citations for these topics are added to the reference section, Ref 101-102.
Round 2
Reviewer 2 Report
I really appreciated the point-by-point responses of the authors. I recommend the pubblication of the manuscript in Molecules.